# NEAR-ZERO-COST DIFFERENTIALLY PRIVATE DEEP LEARNING WITH TEACHER ENSEMBLES

## ABSTRACT

Ensuring the privacy of sensitive data used to train modern machine learning models is of paramount importance in many areas of practice. One approach to study these concerns is through the lens of differential privacy. In this framework, privacy guarantees are generally obtained by perturbing models in such a way that specifics of data used to train the model are made ambiguous. A particular instance of this approach is through a "teacher-student" model, wherein the teacher, who owns the sensitive data, provides the student with useful, but noisy, information, hopefully allowing the student model to perform well on a given task without access to particular features of the sensitive data. Because stronger privacy guarantees generally involve more significant noising on the part of the teacher, deploying existing frameworks fundamentally involves a trade-off between utility and privacy guarantee. One of the most important techniques used in previous work involves an ensemble of teacher models, which return information to a student based on a noisy voting procedure. In this work, we propose a novel voting mechanism, which we call an Immutable Noisy ArgMax, that, under certain conditions, can bear very large random noising from the teacher without affecting the useful information transferred to the student. Our mechanisms improve over the state-of-the-art methods on all measures, and scale to larger tasks with both higher utility and stronger privacy ($\epsilon \approx 0$).

## 1 INTRODUCTION

Recent years have witnessed impressive breakthroughs of deep learning in a wide variety of domains, such as image classification (He *et al.*, 2016), natural language processing (Devlin *et al.*, 2018), reinforcement learning (Silver *et al.*, 2017), and many more. Many attractive applications involve training models using highly sensitive data, to name a few, diagnosis of diseases with medical records or genetic sequences (Alipanahi *et al.*, 2015), mobile commerce behavior prediction (Yan, 2017), and location-based social network activity recognition (Gong *et al.*, 2018). However, recent studies exploiting privacy leakage from deep learning models have demonstrated that private, sensitive training data can be recovered from released models (Nicolas Papernot, 2017). Therefore, privacy protection is a critical issue in this context, and thus developing methods that protect sensitive data from being disclosed and exploited is of paramount importance.

In order to protect the privacy of the training data and mitigate the effects of adversarial attacks, various privacy protection works have been proposed in the literature (Michie *et al.*, 1994; Nissim *et al.*, 2007; Samangouei *et al.*, 2018; Ma *et al.*, 2018). The "teacher-student" learning framework with privacy constraints is of particular interest here, since it can provide a private student model without touching any sensitive data directly (Hamm *et al.*, 2016; Pathak *et al.*, 2010; Papernot *et al.*, 2017). The original purpose of a teacher-student framework is to transfer the knowledge from the teacher model to help train a student to achieve similar performance with the teacher. To satisfy the privacy-preserving need, knowledge from the teacher model is carefully perturbed with random noise, before being passed to the student model. In this way, one hopes that an adversary cannot ascertain the contributions of specific individuals in the original dataset even they have full access to the student model. Using the techniques of differential privacy, such protection can be guaranteed in certain settings. However, the current teacher-student frameworks (e.g. Nicolas Papernot (2017) and (Papernot *et al.*, 2018)) involve a trade-off between student's performance and privacy. This

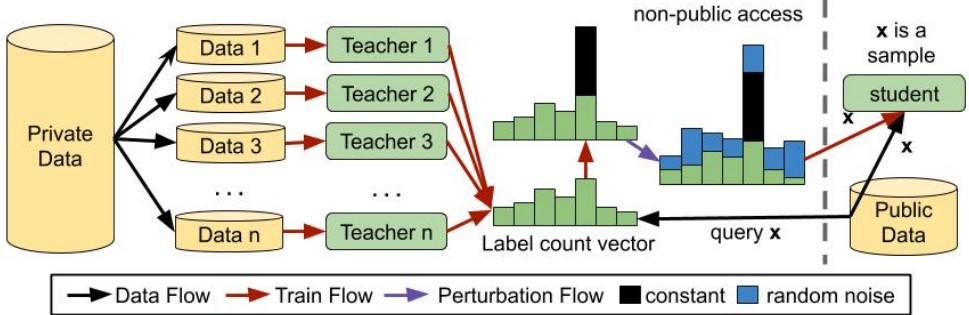

Figure 1: Overview of the proposed approach: firstly, the private data is partition to non-overlapping splits to train the teacher models. To train the student model, the ensemble of teachers then aggregate their predictions on the queried example from student, followed by adding a large constant on the highest counted class vote. The count vector then gets randomly perturbed with noise followed by an ArgMax operation. Finally, the student model is trained by using the returned label from the teacher ensemble.

is because the amount of noise perturbation required is substantial to ensure privacy at the desired level, which leads to degraded information passed to student and results in inaccurate models.

In this paper, we develop a technique to address the aforementioned problem, which facilitates the deployment of accurate models with near zero privacy cost. Instead of using traditional noisy ArgMax, we propose a new approach named immutable noisy ArgMax as describe in Section 2. We redesign the aggregation approach via adding a large constant into the current largest count of the count vector, which enables immutable noisy ArgMax into teacher-student model. As a result, this method improves both privacy and utility over previous teacher-student based methods.

The primary technical contributions of this paper is a novel mechanism for aggregating teachers' answers that are more immutable against larger noise without changing the consensus of the teachers. We show that our proposed method improves the performance of student model on all measures.

## 2 PRELIMINARY

In this section, we briefly overview some background related to derive our new methods. We first introduce some basics in differential privacy, followed by the ArgMax and noisy ArgMax mechanism.

### 2.1 DIFFERENTIAL PRIVACY

To satisfy the increasing demand for preserving privacy, differential privacy (DP) (Dwork *et al.*, 2006) was proposed as a rigorous principle that guarantees provable privacy protection and has been extensively applied in both industrial and academic settings (Andrés *et al.*, 2013; Friedman and Schuster, 2010; Team, 2017).

Let $f$ be a deterministic function that maps the dataset $D$ to the real numbers $\mathbb{R}$. This deterministic function $f$, under the context of differential privacy, is called a query function of the dataset $D$. For example, the query function may request the mean of a feature in the dataset, the gender of each sample. The goal in privacy is to ensure that when the query function is applied on a different but close dataset $D'$, the outputs of the query function are indistinguishable comparing to that from the dataset $D$ such that the private information of individual entries in the dataset can not be inferred by malicious attacks. Here, two datasets $D$ and $D'$ are regarded as adjacent datasets when they are identical except for one single item.

Informally speaking, a randomized data release mechanism for a query function $f$ is said to ensure DP if "neighboring" inputs induce similar distributions over the possible outputs of the mechanism. DP is particularly useful for quantifying privacy guarantees of queries computed over a database with sensitive entries. The formal definition of differential privacy is given below.

**Definition 1** (Differential Privacy Dwork (2011b, Definition 2.4)). *A randomized mechanism $\mathcal{M}$ is $(\epsilon, \delta)$-differentially private if for any adjacent data $D$, and $D'$, i.e $\|D - D'\|_1 \leq 1$, and any output $Y$ of $\mathcal{M}$, we have*

$$\Pr[\mathcal{M}(D) = Y] \leq e^\epsilon \cdot \Pr[\mathcal{M}(D') = Y] + \delta. \tag{1}$$

If $\delta = 0$, we say that $\mathcal{M}$ is $\epsilon$-differentially private. The parameter $\epsilon$ represents the privacy budget (Dwork, 2011a) that controls the privacy loss of $\mathcal{M}$. A larger value of $\epsilon$ indicates weaker privacy protection.

**Definition 2** (Differential Privacy Dwork (2011b, Definition 2.4)). *A randomized mechanism $\mathcal{M}$ is $(\epsilon, \delta)$-differentially private if for any adjacent data $D$ and $D'$, i.e $\|D - D'\|_1 \leq 1$, and any output $Y$ of $\mathcal{M}$, we have*

$$c(o; \mathcal{M}, aux, D, D') \triangleq \log \frac{\Pr[\mathcal{M}(aux, D) = o]}{\Pr[\mathcal{M}(aux, D') = o]}, \tag{2}$$

The privacy loss random variable $C(\mathcal{M}, aux, D, D')$ is defined as $c(\mathcal{M}(d); \mathcal{M}, aux, D, D')$, i.e. the random variable defined by evaluating the privacy loss at an outcome sampled from $\mathcal{M}(D)$.

From the notion of the DP, we know the sensitivity of the deterministic function $f$ (i.e. a query function) about the dataset is important for designing the mechanism for the query function. For different noise mechanisms, it requires different sensitivity estimation. For example, the $\ell_2$-norm sensitivity $\Delta_2 f$ of the query function $f$ is used for Gaussian mechanism which is defined as $\Delta_2 f = \max_{D,D'} \|f(D) - f(D')\|_2$, where $D$ and $D'$ are two neighboring datasets. For the Laplacian mechanism, it uses the $\ell_1$-norm sensitivity $\Delta f$ for random noise sampling. In essence, when the sensitivity is smaller, it means that the query function itself is not very distinguishable given different datasets.

A general method for enforcing a query function $f$ with the $(\epsilon, \delta)$-differential privacy is to apply additive noise calibrated to the sensitivity of $f$. A general method for conveniently ensuring a deterministic query $f$ to be the $(\epsilon, \delta)$-differential privacy is via perturbation mechanisms that add calibrated noise to the query's output (Dwork and Roth, 2014a; Dwork *et al.*, 2010; Nissim *et al.*, 2007; Duchi *et al.*, 2013).

**Theorem 1** (Dwork and Roth (2014b)). *If the $\ell_1$-norm sensitivity of a deterministic function $f$ is $\Delta f$, we have:*

$$\mathcal{M}_f(D) \triangleq f(D) + Lap(\frac{\Delta f}{\varepsilon}), \tag{3}$$

*where $\mathcal{M}_f$ preserves $(\varepsilon, 0)$-differential privacy, and $Lap(b)$ is the Laplacian distribution with location $0$ and scale $b$.*

**Theorem 2** (Dwork and Roth (2014b)). *If the $\ell_2$-norm sensitivity of a deterministic function $f$ is $\Delta_2 f$, we have:*

$$\mathcal{M}_f(D) \triangleq f(D) + \mathcal{N}(0, \Delta_2 f^2 \sigma^2), \tag{4}$$

*where $\mathcal{N}(0, \Delta_2 f^2 \sigma^2)$ is a random variable obeying the Gaussian distribution with mean 0 and standard deviation $\Delta_2 f \sigma$. The randomized mechanism $\mathcal{M}_f(d)$ is $(\epsilon, \delta)$ differentially private if $\sigma \geq \sqrt{2 \ln(1.25/\delta)}/\epsilon$ and $\epsilon < 1$.*

## 2.2 THE ARGMAX MECHANISM

For any dataset $D = \{(x_k, y_k)\}_{k=1}^n$ The *ArgMax Mechanism* is widely used as a query function when $\boldsymbol{v}(x_k) \in \mathbb{N}^d$ is a vector of counts of the dimension same to the number of classes $d$ for sample $x_k$. This decision-making mechanism is similar to the softmax mechanism of the likelihood for each label, but instead of using the likelihood as the belief of each labels, the ArgMax mechanism uses the counts given by the teacher ensembles. Immediately, from the definition of the ArgMax mechanism, we know that the result given by the ArgMax mechanism is immutable against a constant translation, i.e.

$$\arg\max \boldsymbol{v}(x_k) = \arg\max \hat{\boldsymbol{v}}(x_k, c)$$

$$\hat{\boldsymbol{v}}(x_k, c)_i = \begin{cases} \boldsymbol{v}(x_k)_i & \text{if} \quad i \neq \arg\max \boldsymbol{v}(x_k) \\ \boldsymbol{v}(x_k)_i + c & \text{otherwise} \end{cases}$$

where we use subscript $i$ to index through the vector.

## 2.3 THE NOISY ARGMAX MECHANISM

Now, we want to ensure that the outputs of the given well-trained teacher ensembles are differentially private. A simple algorithm is to add independently generated random noise (*e.g.* independent Laplacian, Gaussian noise, etc.) to each count and return the index of the largest noisy count. This noisy ArgMax mechanism, introduced in (Dwork and Roth, 2014b), is $(\varepsilon, 0)$-differentially private for the query given by the ArgMax mechanism.

## 3 OUR APPROACH

In this section, we introduce the specifics of our approach, which is illustrated in Figure 1. We first show the *immutable noisy ArgMax mechanism*, which is at the core of our framework. We then show how this property of immutable noisy ArgMax can be used in a differential private teacher-student training framework.

## 3.1 THE IMMUTABLE NOISY ARGMAX MECHANISM

One interesting observation from the Noisy ArgMax mechanism is that when the aggregated results from the teachers are very concentrated (i.e. most of the predictions agrees on a certain class) and of high counts (i.e. large number of teachers), the result from the ArgMax will not change even under relatively large random noise. The aforementioned scenario is likely to happen in practice, if all the teacher models have relatively good performance on that task. This observation also hints us that if we can make the largest count much larger than the rest counts, we can achieve immutability with significant noise.

Let's define the data transformer as a function that could convert a dataset into a count vector below:

**Definition 3** (Data Transformer). *Given any dataset $D$, the output of data transformer $f$ is an integer based vector, such as $f(D) \in \mathbb{Z}^{|r|}$, where $r$ is the dimension of the vector.*

**Definition 4** (Distance-$n$ Data Transformer). *Given a dataset $D$ and data transformer function $f$, the distance $n$ means the difference between the first and second largest counts given by the $f(D)$ is larger than $n$.*

Then we have the following theorem for immutable noisy ArgMax,

**Lemma 1.** *[Noisy ArgMax Immutability] Given any dataset $D$, fixed noise perturbation vector and a transformer function $f$, the noisy argmax of both $f(D)$ is immutable while we add a sufficiently large constant $c$ into the current largest count of $f(D)$.*

**Theorem 3.** *[Differential private with Noisy ArgMax Immutability] Given any dataset $D$, its adjacent dataset $D'$, fixed noise perturbation vector and a transformer function $f$, while $\Delta f = 1$ ( or $\Delta_2 f = 1$) and the distance of $f(D)$ is larger than 2, the noisy argmax of both $f(D)$ and $f(D')$ is immutable and the same while we add a sufficiently large constant $c$ into the current largest count.*

The proof of the above theorems are provided in the appendix. In essense, when fed with a neighboring dataset $D'$, if the counts of $f(D)$ is different by $n$, the output of the ArgMax mechanism remains unchanged. This *immutability* to the noise or difference in counts due to the neighboring datasets, makes the output of the teacher ensemble unchanged, and thus maintain the advantage of higher performance in accuracy using the teacher ensembles.

## 3.2 NEAR-ZERO-COST QUERY FRAMEWORK

Now, we are ready to describe our near-zero-cost (NZC) query framework.To protect the privacy of training data during learning, NZC transfers knowledge from an ensemble of teacher models trained on non-overlapping partitions of the data to a student model. Privacy guarantees may be understood intuitively and expressed rigorously in terms of differential privacy. The NZC framework consists of three key parts: (1) an ensemble of $n$ teacher models, (2) an aggregation and noise perturbation and (3) training of a student model.

**Ensemble of teachers:** In the scenario of teacher ensembles for classification, we first partition the dataset $D = \{(x_k, y_k)\}_{k=1}^n$ into disjoint sub datasets $\{D_i\}$ and train each teacher $P_i$ separately on each set, where $i = 1, \cdots, t$, $n$ is the number of the dataset and $t$ is the number of the teachers.

**Aggregation and noise perturbation mechanism:** For each sample $x_k$, we collect the estimates of the labels given by each teacher, and construct a count vector $\boldsymbol{v}(x_k) \in \mathbb{N}^L$, where each entry $\boldsymbol{v}_j$ is given by $\boldsymbol{v}_j = |\{P_i(x_k) = j; \forall i = 1, \cdots, t\}|$. For each mechanism with fixed sample $x$, before adding random noise, we choose to add a sufficiently large constant $c$, then we have a new count vector $\hat{\boldsymbol{v}}(x, c)$. Our motivation is not protect $x$ from student, but protect the dataset $D$ from the teacher. Basically, if we fix the partition, teacher training and a query $x$ from student, then we have data transformer $f$ that transfer the target dataset $D$ into a count vector. To be more clear, $x$ and a constant $c$ is used to define the data transformer $f_{x,c}$ and if we query $T$ times, then we have $T$ different data transformer based on each query $x$. Then, by using a data transformer, we can achieve a count vector $\hat{\boldsymbol{v}}(x, c) = f_{x,c}(D)$.

Note that, we uses the following notation that $\hat{\boldsymbol{v}}(x, c)$, also shorted as $\hat{\boldsymbol{v}}$, denotes the data transformer with adding a sufficiently large constant on the largest count, and $\boldsymbol{v}(x)$ denotes the count vector before adding a sufficiently large constant.

We add Laplacian random noise to the voting counts $\hat{\boldsymbol{v}}(x, c)$ to introduce ambiguity:

$$\mathcal{M}(x) \triangleq \arg\max\{\hat{\boldsymbol{v}}(x, c) + Lap(\frac{\Delta f}{\gamma})\},$$

where, $\gamma$ is a privacy parameter and $Lap(b)$ the Laplacian distribution with location 0 and scale $b$. The parameter $\gamma$ influences the privacy guarantee, which we will analyse later.

Gaussian random noise is another choice for perturbing $\hat{\boldsymbol{v}}(x, c)$ to introduce ambiguity:

$$\mathcal{M}(x) \triangleq \arg\max\{\hat{\boldsymbol{v}}(x, c) + \mathcal{N}(0, \Delta_2 f^2 \sigma^2)\},$$

where $\mathcal{N}(0, \sigma^2)$ is the Gaussian distribution with mean 0 and variance $\sigma^2$.

Intuitively, a small $\gamma$ and large $\sigma$ lead to a strong privacy guarantee, but can degrade the accuracy of the pre-trained teacher model and the size of each label in the dataset, as the noisy maximum f above can differ from the true plurality.

Unlike original noisy argmax, our proposed immutable noisy argmax will not increase privacy cost with increasing the number of queries, if we choose a sufficiently large constant $c$ and a large random noise by setting a very small $\gamma$ for Laplacian mechanism (or a large $\sigma$ for Gaussian mechanism). Therefore, for each query, it would costs almost zero privacy budget. By utilizing the property of immutable noisy argmax, we are allowed to have a very large number of queries with near zero privacy budget (setting $c \to +\infty$ and a large random noise for the mechanism).

**Student model:** The final step is to use the returned information from the teacher to train a student model. In previous works, due to the limited privacy budget, one only can query very few samples and optionally use semi-supervised learning to learn a better student model. Our proposed approach enables us to do a large number of queries from the student with near zero cost privacy budget overall. Like training a teacher model, here, the student model also could be trained with any learning techniques.

## 4 PRIVACY ANALYSIS

We now analyze the differential privacy guarantees of our privacy counting approach. Namely, we keep track of the privacy budget throughout the student's training using the moments accountant (Abadi et al., 2016). When teachers reach a strong quorum, this allows us to bound privacy costs more strictly.

### 4.1 MOMENT ACCOUNTANT

To better keep track of the privacy cost, we use recent advances in privacy cost accounting. The moments accountant was introduced by (Abadi *et al.*, 2016), building on previous work (Bun and Steinke, 2016; Dwork and Rothblum, 2016; Mironov, 2016). Definition 3.

**Definition 5.** *Let $\mathcal{M} : D \to \mathbb{R}$ be a randomized mechanism and $D, D'$ a pair of adjacent databases. Let aux denote an auxiliary input. The moments accountant is defined as:*

$$\alpha_{\mathcal{M}}(\lambda) = \max_{aux,d,d'} \alpha_{\mathcal{M}}(\lambda; aux, D, D').$$

*where $\alpha_{\mathcal{M}}(\lambda; aux, D, D') = \log \mathbb{E}[exp(\lambda C(\mathcal{M}, aux, D, D'))]$ is the moment generating function of the privacy loss random variable.*

The moments accountant enjoys good properties of composability and tail bound as given in Abadi *et al.* (2016):

*[Composability]*. Suppose that a mechanism $\mathcal{M}$ consists of a sequence of adaptive mechanisms $\mathcal{M}_1, \ldots, \mathcal{M}_k$, where $\mathcal{M}_i : \prod_{j=1}^{i-1} \mathcal{R}_j \times \mathcal{D} \to \mathcal{R}_i$. Then, for any output sequence $o_1, \ldots, o_{k-1}$ and any $\lambda$

$$\alpha_{\mathcal{M}}(\lambda; D, D') \leq \sum_{i=1}^{k} \alpha_{\mathcal{M}_i}(\lambda; o_1, \ldots, o_{i-1}, D, D').$$

where $\alpha_{\mathcal{M}}$ is conditioned on $\mathcal{M}_i$'s output being $o_i$ for $i < k$.

*[Tail bound]* For any $\epsilon > 0$, the mechanism $\mathcal{M}$ is $(\epsilon, \delta)$-differential privacy for

$$\delta = \min_{\lambda} \exp(\alpha_{\mathcal{M}}(\lambda) - \lambda\epsilon).$$

By using the above two properties, we can bound the moments of randomized mechanism based on each sub-mechanism, and then convert the moments accountant to $(\epsilon, \delta)$-differential privacy based on the tail bound.

## 4.2 ANALYSIS OF OUR APPROACH

**Theorem 4** (Laplacian Mechanism with Teacher Ensembles). *Suppose that on neighboring databases $\mathcal{D}, \mathcal{D}'$, the voting counts $\boldsymbol{v}(x, c)$ differ by at most $\Delta f$ in each coordinate. Let $\mathcal{M}_{x,c}$ be the mechanism that reports $\arg\max_j \boldsymbol{v}(x, c) + Lap(\Delta f/\gamma)$ . Then $\mathcal{M}_{x,c}$ satisfies $(2\gamma, 0)$-differential privacy. Moreover, for any $l$, $aux$, $\mathcal{D}$ and $\mathcal{D}'$,*

$$\alpha(l; aux, \mathcal{D}, \mathcal{D}') \leq 2\gamma^2 l(l+1) \tag{5}$$

For each query $x$, we use the aggregation mechanism with noise $Lap(\Delta f/\gamma)$ which is $(2\gamma, 0)$-DP. Thus over $T$ queries, we get $(4T\gamma^2 + 2\gamma\sqrt{2T \ln \frac{1}{\delta}}, \delta)$-differential privacy (Dwork and Roth, 2014a). In our approach, we can choose a very small $\gamma$ for each mechanism with each query $x$, which leads to very small privacy cost for each query and thus a low privacy budget. Overall, we cost near zero privacy budget while $\gamma \to 0$. Note that, $\gamma$ is a very small number but is not exactly zero, and we can set $\gamma$ to be very small that would result in a very large noise scale but still smaller than the constant $c$ that we added in $\hat{v}$. Meanwhile, similar results are also used in PATE (Papernot *et al.*, 2017), but both our work and PATE is based on the proof of (Dwork and Roth, 2014a). Note that, for neighboring databases $D, D'$, each teacher gets the same training data partition (that is, the same for the teacher with $D$ and with $D'$, not the same across teachers), with the exception of one teacher whose corresponding training data partition differs.

The Gaussian mechanism is based on Renyi differential privacy, and details have been discussed in (Papernot *et al.*, 2018). Similar to the Laplacian mechanism, we also get near zero cost privacy budget overall due to setting a large $\sigma$ and an even larger constant $c$.

In the following, we show the relations between constant $c$ with $\gamma$ and $c$ with $\sigma$ in two mechanism while $\Delta f = 1$ (or $\Delta_2 f = 1$) We first recall the following basic facts about the Laplacian and Gaussian distributions: if $\zeta \sim Lap(1/\gamma)$ and $\xi \sim \mathcal{N}(0, \sigma^2)$, then for $c > 0$,

$$\Pr(|\zeta| \geq c) = e^{-\gamma c}$$

and

$$\Pr(|\xi| \geq c) \leq 2e^{\frac{-c^2}{2\sigma^2}}.$$

Now if each $|\zeta_j| < c$ (resp. $|\xi_j| < c$) for $j = 1, ..., L$, then the $\arg\max$ will not change. We can apply a simple union bound to get an upper bound on the probability of these events.

$$\Pr(\max_{j=1,...,L} |\zeta_j| \geq c) \leq Le^{-\gamma c}$$

and

$$\Pr(\max_{j=1,...,L} |\xi_j| \geq c) \leq 2Le^{\frac{-c^2}{2\sigma^2}}.$$

Thus to obtain a failure probability at most $\tau$, in the Laplacian case we can take $c = \frac{1}{\gamma}\log(L/\tau)$, and in the Gaussian case we can take $c = \sqrt{2\sigma^2\log(2L/\tau)}$.

## 5 EXPERIMENTS

In this section, we evaluate our proposed method along with previously proposed models.

### 5.1 EXPERIMENTAL SETUP

We perform our experiments on two widely used datasets on differential privacy: SVHN Netzer *et al.* (2011) and MNIST LeCun *et al.* (1998). MNIST and SVHN are two well-known digit image datasets consisting of 60K and 73K training samples, respectively. We use the same data partition method and train the 250 teacher models as in Papernot *et al.* (2017). In more detail, for MNIST, we use 10,000 samples as the student dataset, and split it into 9,000 and 1,000 as training and testing set for experiment. For SVHN, we use 26,032 samples as the student dataset, and split it into 10,000 and 16,032 as training and testing set. For both MNIST and SVHN, the teacher uses the same network structure as in Papernot *et al.* (2017).

| Dataset | Aggregator | Queries answered | Privacy bound $\varepsilon$ | Student | Accuracy Clean Votes | Ground Truth |
|---------|-----------|------------------|------------------------------|---------|------------|--------------|
| MNIST | LNMax ($\gamma$=20) | 100 | 2.04 | 63.5% | | |
| | LNMax ($\gamma$=20) | 1,000 | 8.03 | 89.8% | | |
| | LNMax ($\gamma$=20) | 5,000 | > 8.03 | 94.1% | 94.5% | 98.1% |
| | LNMax ($\gamma$=20) | 9,000 | > 8.03 | 93.4% | | |
| | NZC ($c = 1e^{100}, \gamma = 1e^{10}$) | 9,000 | $\approx 0$ | **95.1%** | | |
| | NZC (5 teachers only) | 9,000 | $\approx 0$ | **97.8%** | 97.5% | |
| SVHN | LNMax ($\gamma$=20) | 500 | 5.04 | 54.0% | | |
| | LNMax ($\gamma$=20) | 1,000 | 8.19 | 64.0% | | |
| | LNMax ($\gamma$=20) | 5,000 | > 8.19 | 79.5% | 85.8% | 89.3% |
| | LNMax ($\gamma$=20) | 10,000 | > 8.19 | 84.6% | | |
| | NZC ($c = 1e^{100}, \gamma = 1e^{10}$) | 10,000 | $\approx 0$ | **85.7%** | | |
| | NZC (5 teachers only) | 10,000 | $\approx 0$ | **87.1%** | 87.1% | |

Table 1: Classification accuracy and privacy of the students. LNMax refers to the method from Papernot *et al.* (2017). The number of teachers is set to 250 unless otherwise mentioned. We set $\delta = 10^{-5}$ to compute values of $\varepsilon$ (to the exception of SVHN where $\delta = 10^{-6}$). The baselines refers to student that are trained from the noiseless votes from all teachers. Ground truth refers to student that are trained with ground truth query labels.

| MNIST | | | SVHN | | |
|-------|------|-------|------|------|-------|
| LNMax | NZC | Clean | LNMax | NZC | Clean |
| 93.02% | 94.33% | 94.37% | 87.11% | 88.08% | 88.06% |

Table 2: Label accuracy of teacher ensembles when compared to the ground truth labels from various methods using 250 teachers. Clean denotes the aggregation without adding any noise perturbation.

### 5.2 RESULTS AND DISCUSSIONS

We primarily compare with Papernot *et al.* (2017), which also employs a teacher student framework and has demonstrated strong performance. We did not compare with work from Papernot *et al.* (2018) because the improvements are more on the privacy budget and the improvement of student

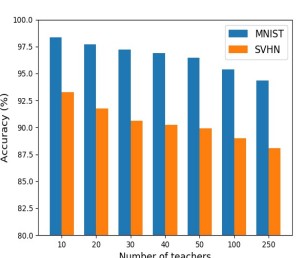
(a) Number of teachers versus Performance

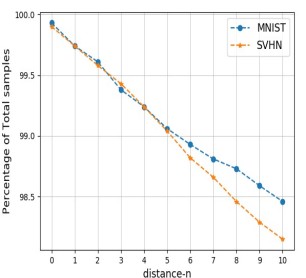
(b) Distance-n versus Qualified Samples with 250 Teachers

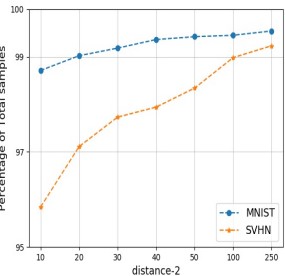
(c) Distance-2 Qualified Samples versus Number of Teachers

Figure 2: (a) shows the trade-off between number of teachers and the performance of the teacher ensemble; (b) shows the percentage of qualified sample which satisfy the distance-n in whole dataset when using 250 teachers; (c) shows the percentage of distance-2 qualified samples over the dataset.

performance on tasks are marginal[1]. We used implementation from the official Github[2], however, we are unable to reproduce the semi-supervised results. Therefore, in the following, we compare the result under fully supervised setting for both approaches.

The results on MNIST and SVHN datasets are shown in table 1. It is clear that the proposed approach achieves both better accuracy and much better privacy cost. In particular, the results are very close to the baseline results, where the student is trained by using the non-perturbed votes from the teacher ensembles. The main reason is that NZC is more robust against the random perturbations for most of the queries, which helps the student to obtain better quality labels for training. We also achieved strong privacy cost, because our approach allows us to use very large noise scale, as long as the constant $c$ is set to a proper large value. To check if the above intuition is true, we calculate the number of correctly labeled queries from the teacher ensembles, and the result is shown in table 2. It is quite clear that our approach is more robust against noise perturbation as compared to the previous approach.

The number of teachers would have a significant impact on the performance of the student, as the teachers were trained on non-overlapping split of the data. The more number of teachers, the less data a teacher has to train. This leads to less accurate individual teachers, and thus less likely to have correct vote for the query. As can be seen from Fig 2a, the performance of the teacher ensembles decreases as the number of teachers increases. This is more prominent for more challenging datasets (e.g. SVHN performance drops more significantly as compared to MNIST). We would like to note that, although the number of qualified samples increases as the number of teachers increase (see Fig 2c), it is at the cost of increasing the wrongly labeled queries, since the total accuracy of teachers has decreased. Because of this, it is likely to result in worse student performance. However, the previous approach such as PATE (Papernot *et al.*, 2017) or Scale PATE (Papernot *et al.*, 2018) requires large number of teachers due to privacy budget constraints. Our approach does not have this limitation. Therefore, we experimented with fewer number of teachers and the results are shown in table 2. The results from using less teachers improved significantly, and approaches closer to the performance when the training student with the ground truth.

## 6 RELATED WORK

Differential privacy is increasingly regarded as a standard privacy principle that guarantees provable privacy protection (Beimel *et al.*, 2014). Early work adopting differential privacy focus on restricted classifiers with convex loss (Bassily *et al.*, 2014; Chaudhuri *et al.*, 2011; Hamm *et al.*, 2016; Pathak *et al.*, 2010; Song *et al.*, 2013). Abadi *et al.* (2016) proposed DP-SGD, a new optimizer by carefully

---

[1]The open source implementation given in https://github.com/tensorflow/privacy only generates table 2 in the original paper from Papernot *et al.* (2018), which does not provide any model performance.

[2]https://github.com/tensorflow/privacy

adding random Gaussian noise into stochastic gradient descent for privacy-preserving for deep learning approaches. At each step of DP-SGD by given a set random of examples, it need to compute the gradient, clip the $l_2$ norm of each gradient, add random Gaussian noise for privacy protection, and updates the model parameters based on the noisy gradient.

Intuitively, DP-SGD could be easily adopted with most existing deep neural network models built on the SGD optimizer. Based on DP-SGD Agarwal *et al.* (2018) applies differential privacy on distributed stochastic gradient descent to achieve both communicate efficiency and privacy-preserving. McMahan *et al.* (2017) applies differential privacy to LSTM language models by combining federated learning and differential private SGD to guarantee user-level privacy.

Papernot *et al.* (2017) proposed a general approach by aggregation of teacher ensembles (PATE) that uses the teacher models' aggregate voting decisions to transfer the knowledge for student model training. Our main framework is also inspired by PATE with a modification to the aggregation mechanism. In order to solve the privacy issues, PATE adds carefully-calibrated Laplacian noise on the aggregate voting decisions between the communication. To solve the scalability of the original PATE model, Papernot *et al.* (2018) proposed an advanced version of PATE by optimizing the voting behaviors from teacher models with Gaussian noise. PATE-GAN Jordon *et al.* (2018) applies PATE to GANs to provide privacy guarantee for generate data over the original data. However, existing PATE or Scale PATE have spend much privacy budget and train lots of teacher models. Our new approach overcomes these two limitations and achieved better performance on both accuracy and privacy budget. Compared with PATE and our model, DP-SGD is not a teacher-student model.

## 7 CONCLUSION

We propose a novel voting mechanism – the immutable noisy ArgMax, which enables stable output with tolerance to very large noise. Based on this mechanism, we propose a simple but effective method for differential privacy under the teacher-student framework. Our method benefits from the noise tolerance property of the immutable noisy ArgMax, which leads to near zero cost privacy budget. Theoretically, we provide detailed privacy analysis for the proposed approach. Empirically, our methods outperforms previous methods both in terms of accuracy and privacy budget.

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

## A  APPENDIX

**Lemma 1.** *[Noisy ArgMax Immutability] Given any dataset $D$, fixed noise perturbation vector and a transformer function $f$, the noisy argmax of both $f(D)$ is immutable while we add a sufficiently large constant $c$ into the current largest count of $f(D)$.*

*Proof.* First, let us recall some facts discussed in the main paper. We first recall the following basic facts about the Laplacian and Gaussian distributions: if $\zeta \sim Lap(1/\gamma)$ and $\xi \sim \mathcal{N}(0, \sigma^2)$, then for $c > 0$,

$$\Pr(|\zeta| \geq c) = e^{-\gamma c}$$

and

$$\Pr(|\xi| \geq c) \leq 2e^{\frac{-c^2}{2\sigma^2}}.$$

Now if each $|\zeta_j| < c$ (resp. $|\xi_j| < c$) for $j = 1, ..., L$, then the $\arg\max$ will not change. We can apply a simple union bound to get an upper bound on the probability of these events.

$$\Pr(\max_{j=1,...,L} |\zeta_j| \geq c) \leq Le^{-\gamma c}$$

and

$$\Pr(\max_{j=1,...,L} |\xi_j| \geq c) \leq 2Le^{\frac{-c^2}{2\sigma^2}}.$$

Thus to obtain a failure probability at most $\tau$, in the Laplacian case we can take $c = \frac{1}{\gamma} \log(L/\tau)$, and in the Gaussian case we can take $c = \sqrt{2\sigma^2 \log(2L/\tau)}$.

Since we have a sufficiently large constant $c$, $c >> \sqrt{2\sigma^2 \log(2L/\tau)}$ or $c >> \frac{1}{\gamma} \log(L/\tau)$, then $c$ minus any sampled noise from either Gaussian or Laplacian distribution is larger than 0 with $1 - \tau$ probability, where we could set $\tau$ as a very small number which is close to 0. Then, the largest count of $f(D)$ adds a positive number which not change the argmax result. □

**Theorem 3.** *[Differential private with Noisy ArgMax Immutability] Given any dataset $D$, its adjacent dataset $D'$, fixed noise perturbation vector and a transformer function $f$, while $\Delta f = 1$ ( or $\Delta_2 f = 1$) and the distance of $f(D)$ is larger than 2, the noisy argmax of both $f(D)$ and $f(D')$ is immutable and the same while we add a sufficiently large constant $c$ into the current largest count.*

*Proof.* First, we have $\Delta f = 1$ (or $\Delta_2 f = 1$) and the distance of $f(D)$ is larger than 2. For any adjacent $D'$, $\arg\max f(D) = \arg\max f(D')$ is immutable, since $f(D')$ can only modify 1 count due to the $\Delta f = 1$. However, the distance is larger than 2, then any modification of $f(D')$ would not change the argmax. Assume the argmax will be changed, let us use $f(D)_{j*}$ presents the largest count and $f(D)_j$ presents the second largest count:

$$f(D)_{j*} - 1 < f(D)_j + 1,$$
$$f(D)_{j*} - f(D)_j < 2,$$

which is conflict the distance-2 of $f(D)$ for any cases.

Then we prove that $f(D)$ and $f(D')$ have the same argmax. By using Lemma 2, we can see that after adding a sufficiently large count and noise perturbation will also not change the argmax information for both $f(D)$ and $f(D')$. Then, we have the same argmax return over any $D$, $D'$ and DP also holds. $\square$

