# OpenReview forum: "Near-Zero-Cost Differentially Private Deep Learning with Teacher Ensembles"
_ICLR.cc/2020/Conference — Reject_

### Official Review · AnonReviewer3 · 2019-10-19
**Official Blind Review #3**

**Rating:** 1

**Review:**

The paper proposes an improvement on the PATE framework for achieving near-zero privacy cost, and showed privacy analyses and experimental evaluations of the proposed method.

The proposed method can be technically flawed. Adding a consent to the max will not guarantee privacy unless you account for the privacy cost for testing whether the distance of f(D) is larger than 2. This is because the distance of f(D) is data-dependent and revealing it violates privacy. Since the whole privacy analysis of PATE is based on the privacy guarantee of the Noisy ArgMax, the epsilon calculated here is voided.

**Experience Assessment:**

I have published in this field for several years.

**Review Assessment: Checking Correctness Of Derivations And Theory:**

I assessed the sensibility of the derivations and theory.

**Review Assessment: Checking Correctness Of Experiments:**

I assessed the sensibility of the experiments.

**Review Assessment: Thoroughness In Paper Reading:**

I read the paper at least twice and used my best judgement in assessing the paper.

---

> ### Author Response · Authors · 2019-11-07
> **Both PATE and our work are data-dependent approach with differential privacy guarantee**
>
> Thank you for reviewing and comments. The following are the reply to some of your comments.
>
> Our work closely follows PATE uses the sensitive data D and partition it into n sub-dataset. Then individual teachers are trained on each data partition. After a student query sample x, then all teachers can return a voting result with the noise perturbation for privacy concerns. It is clear to see that due to the partition policy and uses the partition sub-dataset to train teachers, PATE is a data-dependent approach. Mathematically, PATE has a sensitive data D as input, and need a voting vector output V, the data transformer function f is f_{D, x}(D) -> V, where f is dependent on both sensitive data and student query, then it can transform the sensitive data D to voting vector V. In the whole process of PATE, for each student sample x, the data transformer function f_{D, x} is different, then PATE uses the composition theorem to estimate the total privacy loss of multiple mechanisms.
>
> We follow each step as in PATE, and the difference here is that we use a different function f which adds a constant to the max. This can potentially make the sensitivity among different mechanisms different. However, for each mechanism (i.e. given corresponding query x and dataset D), the sensitivity is fixed and will not be changed. Just like PATE need to know the partition information to bound and estimate the sensitivity of each mechanism, we also need to estimate the sensitivity of each of our mechanisms based on the dataset and the output of the data transformer f_{D, x}. Finally, also use the composition theorem to estimate the total privacy loss of multiple mechanisms. As such, we think our proposed approach is technically correct.
>
> Please let us know if you have any further questions and concerns. Thanks a lot again.

---

### Official Review · AnonReviewer2 · 2019-10-19
**Official Blind Review #2**

**Rating:** 1

**Review:**

This paper studies the teacher ensembles setting for differentially private learning. In this setting, each teacher holds part of the training set and trains a local model. The student uses unlabeled examples to query teacher model. Then the student trains a model from scratch using the examples labeled by teachers.

In order to make the labeling process differentially private, previous work uses noisy argmax mechanism. Each class of label is assigned with a count number. The student first queries the same example to multiple teachers. To guarantee differential privacy, the counts are perturbed by noise before releasing. Then, because of the post-processing property of differential privacy, the argmax operator on such noisy counts are still differentially private.

This paper proposes to add a constant c to the largest count before perturbing and releasing the counts. The authors argue this would improve the accuracy of the noisy argmax operator and yield the same privacy loss as previous approach. However, adding a constant c would increase the sensitivity and therefore degenerates the privacy guarantee. The added noise cannot guarantee the privacy if all others are the same as previous work. To see this clearer, for example, if c=0, then one sample point can at most change the count by 1. If c>0, then one sample point can change the count by 1+c. Because of this, the proposed method cannot guarantee the amount of differential privacy as the paper claimed.


**Experience Assessment:**

I have read many papers in this area.

**Review Assessment: Checking Correctness Of Derivations And Theory:**

I assessed the sensibility of the derivations and theory.

**Review Assessment: Checking Correctness Of Experiments:**

I did not assess the experiments.

**Review Assessment: Thoroughness In Paper Reading:**

I read the paper at least twice and used my best judgement in assessing the paper.

---

> ### Author Response · Authors · 2019-11-07
> **Individual Sensitivity Analysis**
>
> Thank you for reviewing and comments. The following are the reply to some of your comments.
>
> In fact, we mainly follow each step as in PATE. In PATE, for each student sample x, it will create an individual mechanism, and then multiple student data samples lead to multiple individual mechanisms. The difference here is that we propose to use each function f_{D, x} which adds a constant to the max, where D is the sensitive dataset. This can potentially make the sensitivity among different mechanisms different. Then, based on different sensitivities of different mechanisms, we add the noise based on each sensitivity of each individual function.
>
> Since both PATE and our work are data-dependent privacy analysis for each mechanism. In this case, what you mentioned "If c>0, then one sample point can change the count by 1+c" would not be for all mechanisms. Most mechanisms (with different sample points) can still hold the fact that at most change the count by 1 with adding a constant c, instead of c + 1.
>
> Lastly, we estimate the total privacy budget by using the composition theorem. As such, we think our proposed approach is technically correct. Please let us know if you have any further questions and concerns. Thanks a lot again.

---

### Official Review · AnonReviewer1 · 2019-10-24
**Official Blind Review #1**

**Rating:** 1

**Review:**

To improve the privacy-utility tradeoff, this manuscript proposes a voting mechanism used in a teacher-student model, where there is an ensemble of teachers, from which the student can get gradient information for utility improvement. The main idea of the proposed approach is to add a constant C to the maximum count collected from the ensemble, and then noise is furthermore added to the new counts. I can understand that by adding the large constant C, the identity of the maximum count could be preserved with high probability, leading to a better utility on the student side. However, equivalently, this could also be understood as that the noise is not added uniformly across all the counts, but instead a relatively smaller noise is added to the maximum count. Hence it is not clear to me whether the final composition will still be differentially private?


**Experience Assessment:**

I do not know much about this area.

**Review Assessment: Checking Correctness Of Derivations And Theory:**

I did not assess the derivations or theory.

**Review Assessment: Checking Correctness Of Experiments:**

I did not assess the experiments.

**Review Assessment: Thoroughness In Paper Reading:**

I made a quick assessment of this paper.

---

> ### Author Response · Authors · 2019-11-07
> **The final composition is also differential private.**
>
> Thank you for reviewing and comments. The following are the reply to some of your comments.
>
> Yes, our composition theorem is also differential private (DP). In more detail, our work closely follows PATE uses the sensitive data D and partition it into n sub-dataset. Then individual teachers are then trained on each data partition. After a student query sample x, all teachers can return a voting result with the noise perturbation for privacy concerns. Same as in PATE, for each query sample from the student, there is a particular mechanism, and we use the composition theorem to estimate the total privacy loss of multiple mechanisms. It still satisfies the DP requirements. As such, we think our proposed approach is technically correct. Please let us know if you have any further questions and concerns. Thanks a lot again.

---

### Public Comment · ~Ricardo_Silva_Carvalho1 · 2019-09-27
**Not really differentially private**

Hello,

I believe that adding a big constant c depending on who has the largest count does not protect the information of who is the item with largest count, as going from D to D' one would first change the counts (affecting privacy) and only then add the constant. Basically, if on D a given item is the one with the largest count, you're adding the constant to make sure (or with very high prob) such item remains the largest count even after noise addition. That's basically just selecting the item with largest count. And even doing that for a gap > 2 between first/second is not DP.

Having "largest count" minus "second largest count" greater than 2 is not sufficient to satisfy DP, as this property itself is data dependent. The way to use this gap usually is to test privately that D is far from a dataset where the item with largest count is not the first anymore. This is normally done using Propose-Test-Release.
I suggest looking at the papers "Differentially Private Feature Selection via Stability Arguments, and the Robustness of the Lasso" and "Model-Agnostic Private Learning" to see more about this.

So in essence, the way the mechanism is working is revealing information about the item with largest count on D, so not really differentially private.

---

### Decision · Program_Chairs · 2019-12-19

**Decision:**

Reject

**Comment:**

This paper presents a differentially private mechanism, called Noisy ArgMax, for privately aggregating predictions from several teacher models. There is a consensus in the discussion that the technique of adding a large constant to the largest vote breaks differential privacy. Given this technical flaw, the paper cannot be accepted.